# Modeling Human Lung Cells Exposure to Wildfire Uncovers Aberrant lncRNAs Signature

**DOI:** 10.3390/biom13010155

**Published:** 2023-01-12

**Authors:** Piercen K. Nguyen, Yeongkwon Son, Juli Petereit, Andrey Khlystov, Riccardo Panella

**Affiliations:** 1Center for Genomic Medicine, Desert Research Institute, Reno, NV 89512, USA; 2Organic Analytical Laboratory, Division of Atmospheric Sciences, Desert Research Institute, Reno, NV 89512, USA; 3Nevada Bioinformatics Center, University of Nevada Reno, Reno, NV 89557, USA; 4Center for RNA Medicine, Department of Clinical Medicine, Aalborg University, 2450 Copenhagen, Denmark

**Keywords:** lncRNAs, biomarkers, signature, wildfire

## Abstract

Emissions generated by wildfires are a growing threat to human health and are characterized by a unique chemical composition that is tightly dependent on geographic factors such as fuel type. Long noncoding RNAs (lncRNAs) are a class of RNA molecules proven to be critical to many biological processes, and their condition-specific expression patterns are emerging as prominent prognostic and diagnostic biomarkers for human disease. We utilized a new air-liquid interface (ALI) direct exposure system that we designed and validated in house to expose immortalized human tracheobronchial epithelial cells (AALE) to two unique wildfire smokes representative of geographic regions (Sierra Forest and Great Basin). We conducted an RNAseq analysis on the exposed cell cultures and proved through both principal component and differential expression analysis that each smoke has a unique effect on the LncRNA expression profiles of the exposed cells when compared to the control samples. Our study proves that there is a link between the geographic origin of wildfire smoke and the resulting LncRNA expression profile in exposed lung cells and also serves as a proof of concept for the in-house designed ALI exposure system. Our study serves as an introduction to the scientific community of how unique expression patterns of LncRNAs in patients with wildfire smoke-related disease can be utilized as prognostic and diagnostic tools, as the current roles of LncRNA expression profiles in wildfire smoke-related disease, other than this study, are completely uncharted.

## 1. Introduction

Wildfires have been and will remain a huge problem for both environmental and human health and are expected to increase in severity and frequency in several parts of the world [1]. Emissions generated by wildfires are considered one of the most serious threats to public health [2], and it is estimated that 260,000 to 600,000 deaths annually are caused by wildfire smoke exposure [2]. Each wildfire smoke is characterized by a unique chemical composition that is tightly dependent on the geographic origin of the smoke due to factors such as burn type (smoldering vs. flaming), biomass fuel composition, atmospheric aging, and interaction with urban pollutants [3,4]. However, the differential respiratory health impacts induced by unique smokes are largely understudied, and most conclusions have been epidemiologic [5] or focused on tobacco smoke [6].

Classic methods to simulate in vitro wildfire smoke exposure consist of treating lung tissue cultures with liquid wildfire smoke extracts [7]. However, the use of wildfire smoke extracts poses several limitations, as the whole process consists of collecting smoke components using a filter and then extracting them using organic solvents that change particle chemical composition and agglomeration state. Additionally, the exact exposure dosage cannot be determined, and the exposure dose is delivered as a bolus, whereas during inhalation, particles are distributed evenly over a defined period [7]. This method also only extracts soluble compounds and therefore underestimates wildfire smoke as a complex mixture [8].

The 3 Rs (replacement, reduction, and refinement) are a series of principles that serve as the ethical framework for animal research and are designed to improve animal welfare throughout the scientific community [9]. One of the main principles stated in 3R is that the use of animals in studies should be avoided whenever possible [10]. Studies have employed the use of an air-liquid interface (ALI) system to expose in vitro or ex vivo cultures to aerosolized wildfire smoke generated in a burning chamber without the use of an extraction process. The air-liquid interface therefore allows for modeling lung tissue exposure to wildfire smoke without the need for animal models [11]. This therefore supports 3R principles by reducing the number of animals needed to conduct investigations in wildfire smoke toxicology. While scientists are defining specific methodologies for ALI as well as clear standardization and criteria for system validation, the consensus is that ALI provides a more advanced approach to in vitro exposure than classic methods [12].

Little research has been conducted to comparatively examine the effects of exposure to wildfire smoke generated by different fuel types on the unique biological responses in human and mammalian cells. We hypothesized that the chemical composition of wildfire smoke is a critical factor in determining the differential biological effects induced in human lung cells by exposure to unique fuels representative of geographic regions. Moreover, we generated a comprehensive ALI wildfire smoke exposure protocol that accounts for alterations in smoke composition caused by atmospheric aging as smoke travels from the site of origin to the site of exposure in humans and animals. Additionally, our protocol more closely mimics a real-life exposure compared to more classic methods that have all the limitations discussed in the previous section.

Long non-coding RNAs (lncRNAs) are a class of RNA molecules that are longer than 200 nucleotides [13]. They do not contain open reading frames and therefore cannot directly generate peptides or proteins [14]. lncRNAs are the second most abundant class of non-coding RNAs, following pseudogenes [15]. They execute their biological functions with a variety of different mechanisms. They can act as scaffolds to facilitate protein-protein interactions, or directly interact with DNA via specific consensus sequences, through steric 3D structures that repress or enhance the transcription of specific genes. lncRNAs are also reported to interact with mRNA and play a crucial role in the post-transcriptional regulation of several genes [16]. lncRNAs are a very heterogeneous class of RNAs, and even if their roles in regulating the critical aspects of cellular life are now well established and accepted, the molecular biology details of most members of this class are understudied or not studied at all. However, several research groups have begun to shed light on their biogenesis and function. In the past decade, it has been proven that lncRNAs have critical roles in various biological processes, such as inflammation, cell growth, oxidative stress, and apoptosis. For example, lncRNA HIX003209 plays a key role in promoting inflammation in patients with rheumatoid arthritis by sponging miR-6089 via the Tool like receptor 4 (TLR4)/NF-κB signaling pathway [17], and the lncRNA pseudogene PTEN 1 (pg1PTEN) regulates the mRNA of a well-described tumor suppressor gene with apoptotic and anti-proliferative activity [18]. Previous research has also identified lncRNAs as being dysregulated in a variety of human diseases, and they are known to functionally interact with environmental factors such as cigarette smoke [19,20,21]. Moreover, lncRNAs present condition-specific expression patterns that are increasingly being implemented as prognostic and diagnostic biomarkers for human diseases. For example, a three-lncRNA signature accurately predicts the survival of patients with esophageal squamous cell carcinoma (ESCC) [22], and measurement of the lncRNA prostate cancer antigen 3 gene (PCA3) in patient urine samples is an effective method for the diagnosis of early prostate cancer [23]. Additionally, lncRNAs are being investigated as potential diagnostic biomarkers for chronic respiratory diseases such as asthma and chronic obstructive pulmonary disease (COPD) [24].

While several areas of research are currently focused on understanding the role of lncRNAs in tobacco or traffic air-pollution-related respiratory disease, little to no research has been conducted to investigate the response of lncRNAs to wildfire smoke exposure. Here, we utilized a new ALI direct exposure system that we designed and produced in house to prove that exposure to two unique wildfire smokes representative of geographic regions (Sierra Forest and Great Basin) causes aberrant lncRNA expression profiles in a human tracheobronchial cell line (AALE).

## 2. Materials and Methods

### 2.1. Cell Culture Conditions and Seeding

Human immortalized bronchiotrachial cells (AALE) were obtained from the American Type Culture Collection (Manassas, VA, USA. Cells were cultured in Dulbecco’s Modified Eagle Medium (DMEM) supplemented with 10% fetal bovine serum (FBS) and 100 units/mL penicillin in standard cell culture conditions [25].

AALE were counted in a 1:1 dilution of 0.4% trypan blue using a brightfield cell counter 24 h prior to direct smoke exposure (DSE) (DeNovix Celldrop, Wilmington DE, USA), and 1 × 10^6^ cells were replated in 0.35 cm Nunclon-treated cell culture dishes (Thermo Scientific, Waltham, MA, USA, cat. no. 150460) to be incubated in standard cell culture conditions overnight. 10–60 min prior to exposure, the dishes containing cells were transported on ice to the site of the ALI exposure system, and the dishes were placed within the smoke exposure chamber.

### 2.2. Smoke Generation

Wildfire smoke using two different biomass fuels was generated using the Desert Research Institute (DRI) combustion facility under controlled conditions. The facility consists of a burning chamber of aluminum panels (1.83 m × 1.83 m × 2.06 m), an exhaust pipe with multiple sampling ports, and an air inlet at the bottom of the chamber. For each experiment, we burned 150 ± 20 g of biomass fuel. We used Sierra Forest (Smoke A) and Great Basin (Smoke B) fuels to represent the US Western regional fuel characteristics. The Sierra Forest fuel consists of Jeffrey Pine, Ponderosa Pine, Douglas fir, California incense-cedar, Manzanita, and oak, which are found in the Sierra Nevada Mountain region across California and Nevada. The Great Basin region stretches across the western US. Sagebrush, rabbitbrush, antelope bitterbrush, Nevada Mormon tea, pinyon, juniper, and native grasses are commonly found in this region. Each species was collected in appropriate proportions to capture vegetation variability in the regions.

### 2.3. Direct Smoke Exposure (DSE) Using Air-Liquid Interface (ALI) Chamber

The generated smokes were collected within a 1.8 m^3^ Teflon smoke chamber for the direct smoke exposure experiment described in the below section. AALE cells plated as described in Section 2.1 were placed inside the custom-built ALI direct smoke exposure chamber shown in Figure 1c. During exposure, the chambers were placed in a water-bead bath at 37 °C.

### 2.4. Chemical Analysis

We simultaneously collected the Teflon Impregnated Glass Fiber (TIGF) filter (Pall Corporation, Port Washington, NY, USA) followed by the XAD-4 resin (Sigma-Aldrich, St. Louis, MI, USA) cartridge samples for polycyclic aromatic hydrocarbons (PAHs) analysis. The TIGF filter and XAD resin samples were sealed in antistatic zip-lock bags and stored at –20 °C until analysis. After spiking deuterated internal PAH standards, the filters and XAD resin samples were extracted separately with dichloromethane and acetone using an accelerated solvent extractor (ASE) instrument (DIONEX, ASE350, Salt Lake City, UT, USA), then solvents were replaced with toluene (200 µL) for gas chromatography (GC) mass spectrometer (MS) analysis. A Varian CP-3800 GC equipped with a CP-8400 autosampler and interfaced to a Varian 4000 Ion Trap Mass Spectrometer (Varian, Inc., Walnut Creek, CA, USA) was used to perform splitless injections onto a DB-5MS capillary column (30 m, Agilent Technologies, Folsom, CA, USA).

### 2.5. Cellular Growth Assay

AALE were fixed in 10% formalin, washed once with PBS, and stained with 0.1% crystal violet solution (Sigma-Aldrich, St. Louis, MI, USA, prod. no. C0775) 24 h following DSE. Following completion of staining, crystal violet dye was solubilized in 500 µL of 10% acetic acid (Fischer Scientific, Waltham, MA, USA, cat. no. AC124040250) overnight, and the absorbance (OD) of the resulting solution was measured in triplicate at 595 nm using a microplate reader (Spectramax iD5, Molecular Devices, San Jose, CA, USA). Absorbance values were normalized to the non-treated samples.

### 2.6. RNA Extraction

Cells were lysed in 400 uL of TRIzol Reagent (Invitrogen, Waltham, MA, USA cat no. 15596026), and RNA purification including DNAse treatment was conducted using a Direct-zol RNA miniprep (Zymo Research, Irvine CA, USA cat no. R2050S) kit according to the manufacturer’s protocol 8 h following DSE. Purified RNA was eluted in 50 uL of RNAse/DNAse-free water, and 260 nm/230 nm absorbance ratios were confirmed to be greater than 1.5 using a spectrophotometer (Nanodrop One, Waltham, MA, USA).

### 2.7. Library Preperation and Sequencing

RNA was submitted to the Nevada Genomics Core, where libraries were prepared for sequencing using the QuantSeq 3′ mRNA-Seq Library Prep Kit (Lexogen, Vienna, Austria) based on the provided manufacturer instructions and sequenced on an Illumina NextSeq 2000 (Illumina, San Diego, CA, USA) with the P2 100 cycle kit, (Illumina, San Diego, CA, USA) as single reads (SE).

### 2.8. Quality Control and Read Mapping

Sequence data (fastq files) were processed by the Nevada Bioinformatics Core (RRID: SCR_017802). The quality of sequencing reads from the 3′ mRNA-seq was assessed using FastQC v0.11.9 [26] for each sample pre- and post-trimming, and a unified multi-sample report was generated with MultiQC v1.11 [27]. Adapter and primer sequences were removed, and trimming and filtering were conducted with bbduk v38.90 [28]. Quality sequences were then mapped with STAR v2.7.10a [29] to the human reference genome grch38 from GENCODE [30] with the default parameters. To obtain a count table, reads were then quantified with featureCounts v2.0 [31].

### 2.9. Differential Expression Analysis

The count table was imported into the statistical software R v4.2.1 [32]. Transcripts and genes with insufficient count coverage and gene types other than lncRNA were removed for any further analyses. Changes in the gene expression pattern were evaluated using the DESeq2 pipeline v1.36.0 [33]. Pairwise comparative gene expression analyses of smoke exposure to controls for each smoke type (A and B) were conducted, and the variance stabilized transformed data were used for visualization. The false discovery rate (FDR) was calculated to rectify the *p*-values and only genes with a *q*-value < 0.05 were deemed to be differentially expressed. Further, the distributions of the log2 fold changes were evaluated to determine a data-dependent threshold to imply practical significance. Count values, variance stabilized transformed data, and differential analysis results were summarized (Appendix A).

## 3. Results

### 3.1. Exposure System Setup, System Validation and Optimization, and Smoke Exposure Duration

We designed an innovative and novel interface for ALI. To ensure the functionality of the exposure system, we tested, validated, and optimized it by recording the growth curves of different concentrations of AALE cells seeded within the ALI exposure chamber. We also observed the effects on cellular growth after smoke exposure at various lengths of time and confirmed the sensitivity of the system/method.

### 3.2. Exposure System Setup

A detailed description of the ALI experiment set-up that we designed is provided in the Method section. Briefly, as shown in Figure 1a, we set up a system consisting of a burning chamber, a Tedlar smog bag for collecting wildfire smoke, filter-XAD sampler for polycyclic aromatic hydrocarbons (PAHs) sampling, and an ALI exposure chamber. Our system was able to mimic the difference in chemical composition of smoke generated by different types of fuel (Sierra Forest (Smoke A) and Great Basin (Smoke B) fuels).

Figure 1b shows the fraction (%) of the 10 most abundant PAH compounds from the two different fuels. Detailed PAH results are tabulated in Appendix A. Total PAHs concentration (particle and gas phase) of Smoke A was 297.8 μg/m^3^, which was approximately 1.7 times higher than Smoke B (174.3 μg/m^3^). Smoke A and B show different PAH components in smoke. Gas-phase PAHs in both fuels’ emissions mainly consisted of naphthalene and methylnaphthalene. Anthrone, 1,4,5-trimethylnaphthalene, and phenanthrene were the most abundant in the Smoke A particles, while the Smoke B particles mostly contained 1-methylfluorene and fluorene. For both smoke A and B, carcinogenic PAHs were found in the particle phase. Smoke B had a slightly higher carcinogenic benzo[a]pyrene (BaP) concentration (0.664 μg/m^3^) than smoke A (0.533 μg/m^3^). However, dibenzopyrans (e.g., dibenzo[a,h]pyrene and dibenzo[a,i]pyrene), which had 30–100 times higher carcinogenicity than BaP, in smoke A were much higher than in smoke B. AALE cells were exposed to the two distinct wildfire smokes using the ALI custom designed exposure chambers showed in Figure 1c and detailed in the Materials and Methods section to achieve a direct smoke exposure (DSE).

### 3.3. Validation and Optimization of Cell Seeding within ALI Exposure Chambers

For the validation of cell viability within the ALI exposure chambers and optimization of seeding concentration, AALE were plated at several different concentrations ranging from 5 × 10^3^ to 1 × 10^6^ within the ALI exposure chamber; these were observed, and hours were recorded when 100% confluency was reached. As shown in Figure 1d, AALE cells were viable inside the ALI exposure chamber, and their numbers were in line with the expected growth curves for each seeding concentration. Cells seeded at 1 × 10^6^ cells per mL became 100% confluent within 48 h; therefore, 1 × 10^6^ cells per mL was designated as the optimal seeding concentration.

### 3.4. Validation and Optimization of Smoke Exposure Duration and Experimental Setup

For optimization and validation of exposure time length, AALE were exposed in duplicate to wildfire smoke A for the following lengths of time: non-treated, 5 min, 15 min, 30 min, 60 min, 120 min, and 240 min (an overview of the experiment design is provided in Figure 1e). Results shown in Figure 1f and quantified in Figure 1g demonstrate that cellular growth was dramatically reduced following 120 min of exposure. Therefore, 60 min was identified as the optimal exposure time allowing for workable yield of RNA collection from viable cells.

To test our hypothesis, we designed the experiment presented in Figure 1h, and we proceeded to extract total RNA from AALE cells after 8 h of smoke exposure. Cells were exposed to smokes from the Sierra Forest (Smoke A) or the Great Basin (Smoke B), or no smoke (NT), representing the control samples. Extracted RNA was purified and sequenced to determine how different smokes induce specific transcriptomic changes in human lung epithelial cells.

### 3.5. RNA Seq Analysis of LncRNA

For the nine sequenced samples, we obtained 10.1–15.3 million (M) aligned reads. After filtering, 2326 lncRNAs with good read coverage remained for further analyses. The principal component analysis (PCA) of the variance stabilized transformed data distinctly grouped the experimental conditions, indicating that each smoke has a unique lncRNA expression signature (Figure 2a). This was further strengthened by the differential analysis, which resulted in 26 and 256 differentially expressed (DE) lncRNAs between Smoke A and B compared to control, respectively (Figure 2b, left). In addition to the statistical significance, a practical significance threshold was also employed to determine the lncRNAs with the largest expression changes compared to the control. Applying a log2 fold change (log2 FC) threshold of 6, the number of lncRNAs reduced substantially for the Smoke B comparison to the control; about 17% of DE lncRNAs have at least an absolute log2 FC of 6 in Smoke B compared to the control, whereas 23% of DEs are retained in the Smoke A comparison (Figure 2b, right). The 48 lncRNAs presented in Figure 2b, right, are further visualized in the heatmap (Figure 2c). The heatmap shows that many lncRNAs are suppressed when exposed to Smoke B. Figure 2d presents the results of the differentially analyzed data by plotting the log2 FC against the significance value. Here, a threshold of log2 FC of 2 was used to highlight genes, resulting in 11 and 205 lncRNAs, respectively. These results indicate that smoke A has less of an effect on the healthy lung cells compared to smoke B, although the identified changes are noticeable in both smoke exposures. Furthermore, smoke B has notably more downregulated lncRNAs compared to upregulated lncRNAs, 177 and 28, respectively.

These results further demonstrate that each smoke has a unique effect on healthy lung tissue cells; thus, the health effects caused by wildfires cannot be generalized and depend on the composition of the smoke.

## 4. Discussion

LncRNA signatures are emerging as powerful biomarkers in both prognostic and diagnostic applications. For example, a 6-lncRNA prognostic signature for predicting the prognosis of patients with colorectal cancer metastasis [34]. With the already hazardous air quality resulting from wildfire smoke in several areas of the world projected to increase, it is important to improve the understanding of health outcomes caused by wildfire smoke exposure [35,36]. Being able to recognize unique expression patterns of lncRNAs in patients with wildfire smoke-related disease would provide avenues for healthcare professionals to provide specialized and unique treatments based on improved prognostic and diagnostic tools. However, the current roles and expression patterns of lncRNAs in wildfire smoke-related disease are completely uncharted.

Results obtained from our investigation prove that the effects of wildfire smoke on the cellular transcriptome are unique and strictly linked to the chemical composition of the smoke. Therefore, we demonstrate that the common misconception that wildfire-generated smokes are all equally dangerous for human health is an oversimplification because each smoke can induce specific transcriptomic changes in the same human lung cell line. Moreover, linking the lncRNA expression profiles to a specific type of wildfire smoke provides a rationale for designing a specific panel of lncRNA that can be used to understand which type of wildfire a patient has been exposed to and provides a deeper insight on how smoke from different fuels uniquely affects human health. Our data proves that different types of smoke can have differential effects on respiratory health and provide clues for future studies investigating the roles of lncRNAs in wildfire smoke-related disease. Our study describes for the first time, the lncRNA landscape of a human lung cell line directly exposed to wildfire smoke, and it is the first study to analyze the unique transcriptomic effects induced by multiple wildfire smokes representative of geographic areas. Our results elucidate the importance of chemical composition (and therefore geographic origin) in assessing the respiratory health risks that may result from wildfire smoke exposure. We used the RNA-seq data generated through our experiments to run an unbiased analysis aiming to identify differentially expressed transcripts, and here, through both stringent and global analysis, we demonstrate that different wildfire smokes induce aberrant expression of lncRNAs. Additionally, our results suggest that lung cell lncRNA expression profiles of patients exposed to wildfire smoke may be used to identify, to some degree, the chemical composition and/or geographic origin of the emissions they inhaled. This opens the possibility of lncRNA expression profiles being used in the prognosis and diagnosis of patients facing acute or chronic exposures to wildfire smoke.

Furthermore, this proof-of-concept study, along with the known fact that different lncRNAs have a dramatic differential effect on cellular processes such as inflammation [37], is a very strong rationale to hypothesize that different smokes derived from different fuels can generate a diverse inflammation response in the human lungs, promoting the activation of different pro-inflammatory pathways or the inhibition of anti-inflammatory mechanisms. Dissecting the molecular mechanisms that control different types of inflammation is the first step to design and develop new therapeutic approaches and strategies that can be extremely effective and provide strong benefit to patients affected by wildfire-induced lung toxicity, preventing the inflammation from becoming chronic and preventing further complications that may lead to serious health conditions culminating with lung cancer.

Our observation provides a strong proof of principle, and we recognize that more studies should be conducted to corroborate them. Additional studies should be conducted to investigate fuel types in other global regions heavily impacted by wildland fire emissions. Environmental factors such as photochemical aging, interaction with urban traffic pollutants, and anthropogenic materials should also be investigated to provide a more comprehensive view of wildfire exposure resulting from smoke that has traveled from the site of origin to populated communities. Different cellular models are needed, for example, human bronchial epithelial cells (BEAS-2B) or human non-small cell lung cancer cells (H3255), to understand differential cellular responses to exposure to the same type of smoke. This will elucidate differential respiratory health risks for sensitive populations (patients with lung cancer, asthma, etc.) on a molecular level.

Since we proved that alteration of the cellular transcriptome is fuel-specific, we also speculate that the coding area is affected by the same factors. We foresee that the medical and research fields will soon disentangle the differences in the expression of protein or peptide-coding genes that result from exposure to unique wildfire smoke. The cumulative knowledge that the scientific community can obtain from the perturbation of the transcriptome from both a coding and non-coding perspective will allow patients experiencing illness as a result of wildfire smoke exposure to be treated on an exposure-specific level, accounting for the unique pathological differences of unique smoke types. This will allow for the most rapid and effective treatment of individuals suffering from wildfire smoke-related pulmonary disease.

## Figures and Tables

**Figure 1 biomolecules-13-00155-f001:**
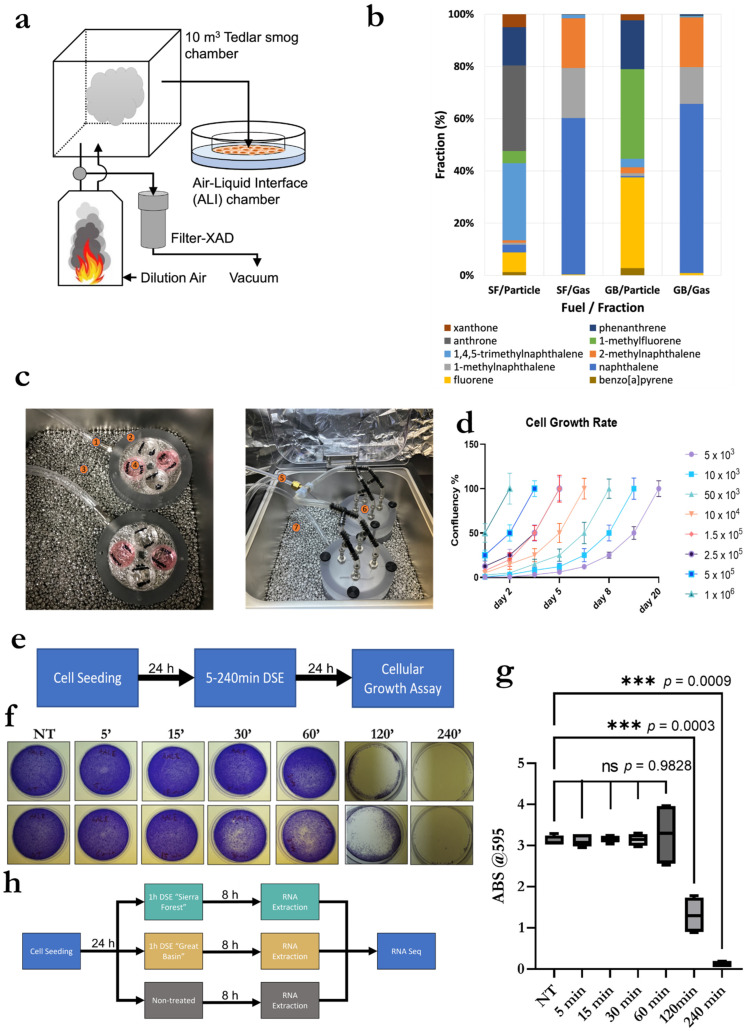
Direct smoke exposure affects cell viability in a time dependent manner. (**a**) Schematics of smoke exposure system; (**b**) Concentrations of most abundant Polycyclic aromatic hydrocarbon (PAH) compounds for two tested fuels; (**c**) In vitro exposure system setup consisting of smoke inlet (1, 5, and 6), ALI chamber housing. (2) In-vitro cell culture (4), and incubator with metal beads (3, 7); (**d**) Cell growth rates of AALE seeded and incubated within the exposure system; (**e**) Graphic representation of experimental design to validate cell viability in the ALI exposure system; (**f**) Crystal Violet assay performed on cells exposed to smoke in the ALI device; (**g**) Quantification of Crystal Violet results with statistical significance *** indicate a statistical significance with a *p* value < 0.001; ns indicates a non-significant statistical difference; (**h**) graphic representation of experimental design for sample collection for RNA seq analysis.

**Figure 2 biomolecules-13-00155-f002:**
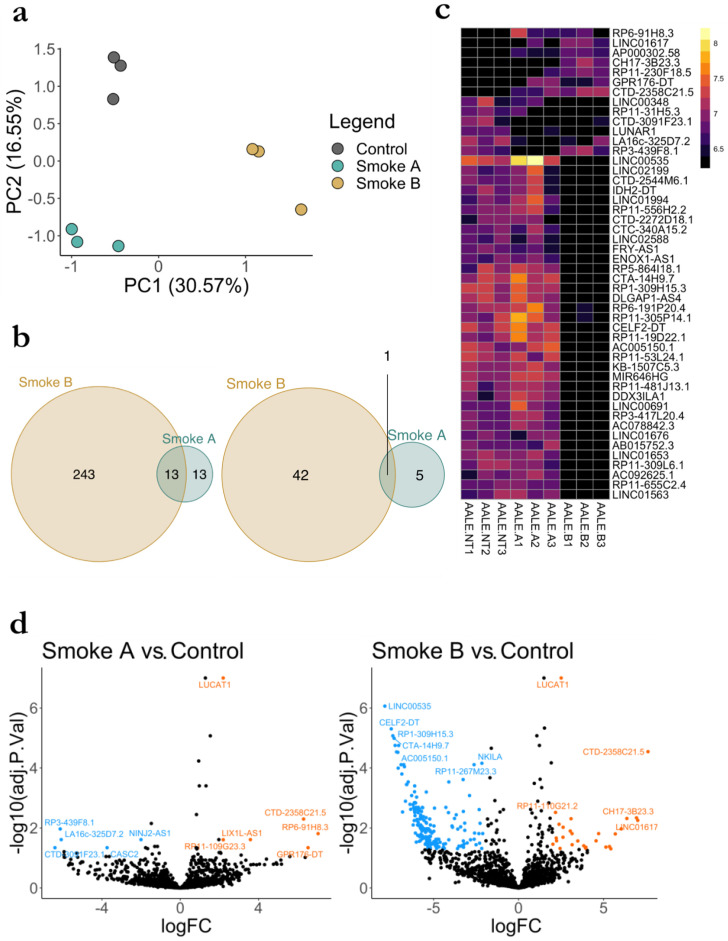
Differential lncRNA signatures correlate with DSE for different fuels. (**a**). Principal component analysis (PCA) of variance stabilized transformed (vst) lncRNAs data, *n* = 2326; (**b**) Venn diagram of differentially expressed lncRNAs; (left) includes all lncRNAs with a false discovery rate adjusted *p*-value < 0.05 (right) is a subset of (left) with an imposed log2 fold change threshold of greater or less than 6 or −6; (**c**). Expression heat map of variance stabilized transformed (vst) lncRNAs data showing the largest changes in expression patterns compared to control (NT), this is the set of lncRNAs in (b-right), *n* = 48; (**d**) Volcano plot of each pairwise comparison to control (NT), highlighted genes are down (blue) or upregulated (orange), with a false discovery rate adjusted *p*-value < 0.05 and a log2 fold change of greater or less than 2 or −2.

## Data Availability

Not applicable.

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
