# Peer review of "Modeling Human Lung Cells Exposure to Wildfire Uncovers Aberrant lncRNAs Signature"

_biomolecules, 2023, doi:10.3390/biom13010155_

Round 1
Reviewer 1 Report
In the present study “Modeling Human Lung Cells Exposure to wildfire Uncovers Aberrant lncRNAs Signature”, Nguyen et al. utilized a newly designed air-liquid interface direct exposure system, to expose human tracheobronchial epithelial cells (AALE) to smokes of different chemical composition.
A preliminary investigation of the effects carried out from AALE-smoke exposure, was performed by monitoring the cell growth rates. RNA-seq analysis was then performed to link and evaluate changes in gene expressions. The authors aim at defining a transcriptome signature that correlates with different nature of smokes.
The study presents for the first time an interesting and innovative system, developed to mimic the condition of cells exposure to wildfire smokes from different geographic area.
Overall, I found the manuscript to be a pleasure to read and I have only minor comments listed below.
Minor Criticisms
· (From lanes 86-86) “Previous research has also identified lncRNAs to be dysregulated in a variety of human diseases and they are known to functionally interact with environmental factors such as cigarette smoke.”
Please, can the authors add references of lncRNAs dysregulated by cigarette smoke? In the specific case of human cells exposed to wildfire smokes and consequent dysregulation of lncRNAs expression, it seems a completely new research topic. I wonder if anything similar was published before the present study. Please can authors verify and in case report some examples?
· Figure 1a. As first impact looking at the scheme, I was confused by the Air Liquid Interface chamber (ALI), which is here schematized as a transparent petri dish. I would suggest to add few details, maybe drawing the four wells or use a different color to individuate were the cells are seeded.
· Figure 1g. There is no graph, probably was accidentally deleted during the pdf formatting, please re-place it.
· Figure 2. Can the authors increase all fonts especially for axis titles, sometimes they are hard to read especially in the printed version of the figures.
Author Response
Response to reviews of the manuscript by Nguyen et al. entitled “Modeling Human Lung Cells Exposure to Wildfire Uncovers Aberrant lncRNAs Signature” (biomolecules-2106083).
We would like to thank the reviewers for their objective and thorough review of our manuscript. Please, find below our detailed response to the reviews. We sincerely believe that we have adequately addressed all the comments raised by the reviewers and thus, hope that the revised manuscript in its present form could be accepted for publication in Biomolecules.
Response to Reviewer #1 comments:
“Previous research has also identified lncRNAs to be dysregulated in a variety of human diseases and they are known to functionally interact with environmental factors such as cigarette smoke.”
Please, can the authors add references of lncRNAs dysregulated by cigarette smoke? In the specific case of human cells exposed to wildfire smokes and consequent dysregulation of lncRNAs expression, it seems a completely new research topic. I wonder if anything similar was published before the present study. Please can authors verify and in case report some examples?
We thank Reviewer for the comment and have addressed it by adding three references to line 102 regarding previous research demonstrating that lncRNAs can be dysregulated by cigarette smoke.
Figure 1a. As first impact looking at the scheme, I was confused by the Air Liquid Interface chamber (ALI), which is here schematized as a transparent petri dish. I would suggest to add few details, maybe drawing the four wells or use a different color to individuate were the cells are seeded.
We appreciate the useful comment from the reviewer. We have edited figure 1a to included more details to individuate where the cells were seeded within the ALI exposure chamber.
Figure 1g. There is no graph, probably was accidentally deleted during the pdf formatting, please re-place it.
We thank the reviewer for noticing the error and have corrected Figure 1g.
Figure 2. Can the authors increase all fonts especially for axis titles, sometimes they are hard to read especially in the printed version of the figures.
We have increased the font size to from 11pt to 12pt and changed the style from Garamond to the more commonly used Arial font.
A Reviewed version of the original manuscript and figures is attached to this response

Reviewer 2 Report
General comment
- The study titled "Modeling Human Lung Cells Exposure to Wildfire Uncovers 2 Aberrant lncRNAs Signature" is highly intriguing since it helps us better understand how wildfires affect lung disease and provides information on the many physiologically regulating molecules known as "lncRNAs."
- Although they require some modifications, as noted in the specific remarks, the abstract, introduction, methods, result, discussion, and conclusions are good and well organized.
Specific comments
Introduction
- Better to rephrase sentence in line 31-33".... the world including the Western U.S.", it is not clear what does it means the western U.S.
- Better to rewrite the sentence " The air liquid interface ...... vitro or ex vivo setting therefore, supporting 3R principles in replacing.... for these studies" Line 51-54
- give explanation about 3R: line 53
- The sentence "it may...... based on fuel type induce unique biological responses....." in line 58-61 needs rewriting. "Based on fuel type induce unique" is not understandable
Result
-It is preferable to split Figure 1 into separate result sections, with the matching figure appearing after the first citation of the figure in the text.
- Figure 1g does not show any quantitative data and nothing say in the result section
- Better to improve the font types of Legends on figure 1 and 2
- " cristal violet" line 147 changed into crystal violet
Discussion
- Better to move the sentence in line 204-205 after the sentence in line 206-208.
Author Response
Response to reviews of the manuscript by Nguyen et al. entitled “Modeling Human Lung Cells Exposure to Wildfire Uncovers Aberrant lncRNAs Signature” (biomolecules-2106083).
We would like to thank the reviewers for their objective and thorough review of our manuscript. Please, find below our detailed response to the reviews. We sincerely believe that we have adequately addressed all the comments raised by the reviewers and thus, hope that the revised manuscript in its present form could be accepted for publication in Biomolecules.
Response to Reviewer #2 comments:
Introduction
-Better to rephrase sentence in line 31-33".... the world including the Western U.S.", it is not clear what does it means the western U.S.
We appreciate the reviewers comment and have edited the sentence to be more concise:
“Wildfires have been and will remain a huge problem for both the environmental and human health and are expected to increase in severity and frequency in several parts of the world3.”
-Better to rewrite the sentence " The air liquid interface ...... vitro or ex vivo setting therefore, supporting 3R principles in replacing.... for these studies"
We agree with the reviewer and have replaced the sentence with the following:
“The air liquid interface therefore allows for modeling lung tissue exposure to wildfire smoke without the need of animal models11. This therefore supports 3R principles by reducing the number of animals needed to conduct investigations in wildfire smoke toxicology.”
-give explanation about 3R: line 53
We thank the reviewer for the useful comment. We have added an explanation to 3R to the manuscript as follows:
“The 3Rs (Replacement, Reduction, and Refinement) are a series of principles that serve as the ethical framework for animal research and are designed to improve animal welfare throughout the scientific community9. One of the main principles stated in 3R is that the use of animals in studies should be avoided whenever possible10”.
-The sentence "it may...... based on fuel type induce unique biological responses....." in line 58-61 needs rewriting. "Based on fuel type induce unique" is not understandable
We agree with the reviewer and have replaced the sentence with the following:
“Few research has been conducted to comparatively examine the effects of exposure to wildfire smoke generated by different fuel types on the unique biological responses in human and mammalian cells.”
Result
-It is preferable to split Figure 1 into separate result sections, with the matching figure appearing after the first citation of the figure in the text.
We thank the reviewer for the suggestion and split Figure 1 into separate result sections.
-Figure 1g does not show any quantitative data and nothing say in the result section
We thank the reviewer for the comment and have corrected the graph and result section:
“Results shown in Figure 1f and quantified in Figure 1g demonstrate that cellular growth was dramatically reduced following 120 min of exposure.”
-Better to improve the font types of Legends on figure 1 and 2
We thank the reviewer for the suggestion and have changed the font types of both figure 1 and 2.
-" cristal violet" line 147 changed into crystal violet
We thank the reviewer for noticing the error and have corrected it.
Discussion
- Better to move the sentence in line 204-205 after the sentence in line 206-208.
We thank the reviewer for the suggestion and have moved the sentences appropriately.
A reviewed version of the manuscript and figures is attached to this response
